# The Analgesic Efficacy of the Single Erector Spinae Plane Block with Intercostal Nerve Block Is Not Inferior to That of the Thoracic Paravertebral Block with Intercostal Nerve Block in Video-Assisted Thoracic Surgery

**DOI:** 10.3390/jcm11185452

**Published:** 2022-09-16

**Authors:** Sujin Kim, Seung Woo Song, Hyejin Do, Jinwon Hong, Chun Sung Byun, Ji-Hyoung Park

**Affiliations:** 1Department of Anesthesiology and Pain Medicine, Wonju College of Medicine, Yonsei University, Wonju 26426, Korea; 2Department of Thoracic and Cardiovascular Surgery, Wonju College of Medicine, Yonsei University, Wonju 26426, Korea

**Keywords:** video-assisted thoracic surgery, erector spinae plane block, paravertebral block, intercostal nerve block

## Abstract

This monocentric, single-blinded, randomized controlled noninferiority trial investigated the analgesic efficacy of erector spinae plane block (ESPB) combined with intercostal nerve block (ICNB) compared to that of thoracic paravertebral block (PVB) with ICNB in 52 patients undergoing video-assisted thoracic surgery (VATS). The endpoints included the difference in visual analog scale (VAS) scores for pain (0–10, where 10 = worst imaginable pain) in the postanesthetic care unit (PACU) and 24 and 48 h postoperatively between the ESPB and PVB groups. The secondary endpoints included patient satisfaction (1–5, where 5 = extremely satisfied) and total analgesic requirement in morphine milligram equivalents (MME). Median VAS scores were not significantly different between the groups (PACU: 2.0 (1.8, 5.3) vs. 2.0 (2.0, 4.0), *p* = 0.970; 24 h: 2.0 (0.8, 3.0) vs. 2.0 (1.0, 3.5), *p* = 0.993; 48 h: 1.0 (0.0, 3.5) vs. 1.0 (0.0, 5.0), *p* = 0.985). The upper limit of the 95% CI for the differences (PACU: 1.428, 24 h: 1.052, 48 h: 1.176) was within the predefined noninferiority margin of 2. Total doses of rescue analgesics (110.24 ± 103.64 vs. 118.40 ± 93.52 MME, *p* = 0.767) and satisfaction scores (3.5 (3.0, 4.0) vs. 4.0 (3.0, 5.0), *p* = 0.227) were similar. Thus, the ESPB combined with ICNB may be an efficacious option after VATS.

## 1. Introduction

Although it is less invasive and damaging to tissue, video-assisted thoracic surgery (VATS) still causes moderate to severe postoperative pain [1]. Postoperative pain is an independent predictor of mortality and morbidity [2]. In addition, acute pain after thoracic surgery is associated with chronic pain; therefore, postoperative pain control is a main goal [3]. The thoracic epidural block (TEB) is the standard procedure for a regional block in thoracotomy; however, the paravertebral block (PVB) is known to show equipotent efficacy [4]. These are also widely used in VATS procedures. The erector spinae plane block (ESPB) is a simple fascial plane block that serves as an alternative to an epidural block, with fewer side effects [5]. It appears to be an effective analgesic technique at many levels and functions as an alternative when the PVB or epidural block is contraindicated [6]. However, some clinical trials have shown that ESPB has a lesser analgesic effect than PVB [7]; thus, no consensus has been reached. The intercostal nerve block (ICNB) is a multimodal analgesia technique following thoracic surgery, and its analgesic efficacy has been proven [8]. However, no study has compared the analgesic effects of ESPB and PVB when combined with ICNB.

We hypothesized that the ESPB would not be less effective than the PVB for attenuating pain when combined with ICNB after VATS. The primary outcome of this monocentric, single-blinded, randomized controlled noninferiority trial was the median difference in postoperative pain scores between the groups. The secondary outcomes were the cumulative postoperative analgesic consumption (calculated as oral morphine milligram equivalents, MME) and patient satisfaction scores.

## 2. Materials and Methods

### 2.1. Study Population

The study trial was approved by the Institutional Review Board of Yonsei University Wonju College of Medicine, Wonju, Korea, and the participants are listed at https://cris.nih.go.kr (accessed on 18 May 2021, KCT 0006271). We enrolled 52 patients with American Society of Anesthesiologists physical status I, II, or III, aged 19–80 years, who had undergone VATS between June 2021 and January 2022. Patients were excluded from the study for cognitive impairment, anticoagulant administration, coagulopathy, antiplatelet drug administration within 48 h, double antiplatelet therapy, surgical site infection, refusal of procedure, allergic reaction to local anesthetic, requiring therapeutic anticoagulant therapy after surgery, or pregnancy. Patients with comorbid diseases were excluded according to the judgment of the anesthesiologist (sepsis, anatomical thoracic deformity, empyema, increased intracranial pressure, etc.). Patients were randomly and evenly assigned to either the PVB with ICNB group or the ESPB with ICNB group by a computer-generated randomization table. Blinding of the group designation was maintained for the patients and attending anesthesiologists, except one practitioner (J.-H.P.) who performed the PVB or ESPB.

### 2.2. Perioperative Management

Anesthesia was induced by a bolus of propofol (1.5–2 mg/kg) and remifentanil (1 mcg/kg). Rocuronium (0.6 mg/kg) was used for tracheal intubation. The remifentanil infusion rate was adjusted by the attending anesthesiologist according to the overall hemodynamic data and the suggested intensity of surgical stimuli. The fraction of inhaled anesthetics was administered under BIS guidance. The surgeon performed the ICNB and placed a chest tube during the surgery. At the end of the surgery, the ESPB or PVB was performed in the lateral position. One anesthesiologist (J.-H.P.) performed the PVB and ESPB. The same experienced surgeon (C.S.B.) performed the ICNB. The intravenous patient-controlled analgesia (PCA) was used at the discretion of the anesthesiologist, and the dose was recorded in terms of morphine milligram equivalents (MME). The patients were extubated and transported to the postanesthetic care unit (PACU). The standard analgesic algorithm in PACU was intravenous nonopioid analgesics for visual analog scale pain scores of 4–6 [1] and intravenous fentanyl (50 µg) for VAS pain scores >6 [2]. The postoperative pain in the ward was controlled by the primary physician. Administered analgesic drugs were converted into MME and recorded in case record forms. The analgesics used in the ward were intravenous tramadol, intramuscular or subcutaneous meperidine, oral Ultracet^®^ (tramadol 37.5 mg/acetaminophen 325 mg), and transdermal fentanyl patch.

### 2.3. Paravertebral Block

After surgery, each patient was placed in the lateral decubitus position, and skin preparation was performed. The patient was palpated at the T5 level, and the linear transducer was positioned in a vertical plane approximately 2.5 cm lateral to the palpated spinous process, obtaining a sagittal plane of the transverse process, superior costovertebral ligament, intertransverse ligaments, desired paravertebral space, pleura, and lung tissue. The paravertebral space was bordered by the vertebral body, intervertebral foramen, parietal pleura, and costovertebral ligament. A 21-gauge, 10 cm echogenic needle (Vygon SA; Ecouen, France) was placed in the paravertebral space using an in-plane approach to confirm that there was no blood aspiration. A small amount of local anesthetic was then injected into the test dose in real time to reduce the anterior displacement of the pleura and spine. Ropivacaine (0.375%, 20 cc) was injected with aspiration every 5 cc to prevent intravascular injection.

### 2.4. Erector Spinae Plane Block

At the T5 level, the linear transducer was moved slowly from the midline to the lateral, and the transducer was moved approximately 3 cm until a transverse projection was observed. It was distinguishable from the ribs at that level: the transverse process is shallow and wide, whereas the ribs are deep and thin. The trapezius, rhomboid, and erector spinae muscles were then identified. An echogenic needle was inserted from the head to the foot, using an in-plane approach, and advanced toward the transverse process through the trapezius, rhomboid, and erector spinae muscles under ultrasound guidance. A small amount of local anesthetic was administered when the needle tip was located below the erector spinae muscle. The correct position of the needle was verified by visually confirming that the erector spinae muscle was separated from the transverse process. Ropivacaine (0.375%, 20 cc) was injected with aspiration every 5 cc to prevent intravascular injection.

### 2.5. Intercostal Nerve Block

At the end of the surgical procedure, a total of 10 cc of ropivacaine (2 cc per space) was injected into the intercostal space until swelling of the intercostal nerve at the T4–T8 levels occurred.

### 2.6. Outcome Measures

The primary endpoint of the present study was to assess the analgesic efficacy of ESPB compared with that of PVB when combined with ICNB by measuring the median differences between the groups in the VAS of pain at the PACU, as well as 24 and 48 h after surgery. The secondary endpoints were to investigate the total amount of analgesics administered to the patients in MME and the satisfaction score of patients using a five-point rating scale.

### 2.7. Sample Size Calculation

The standard deviation for PVB with ICNB in the pilot study was 2.29, and no significant difference was observed in the variance between the ESPB and PVB in a previous study [9]. When the noninferiority margin (delta value) was set to 2, it was determined based on the opinion of colleagues and a previous study [10]. The significance level was set to 0.05, and the power was set to 0.9; accordingly, the estimated number of patients required in each group was 23. Accounting for a dropout rate of 10%, we decided to enroll 26 patients in each group.

### 2.8. Statistical Analysis

All statistical analyses were performed using the IBM SPSS statistical software package (IBM SPSS Statistics for Windows, version 25, IBM Corporation, Armonk, NY, USA). Distribution of continuous variables was assessed using the Shapiro–Wilk test. Intergroup comparisons of the non-normally distributed variables were performed using the Mann–Whitney U test and are reported as the median (interquartile range). Intergroup comparisons of other variables that showed a normal distribution were tested using an independent *t*-test and are reported as the mean ± standard deviation (SD). For pain scores assessed at three timepoints, a post hoc Bonferroni correction was applied to adjust for multiple comparisons.

## 3. Results

In total, 52 patients were screened, and all of them were enrolled and assigned to the two groups. There was no dropout among the enrolled patients. Hence, all 52 patients were included in the final analysis.

The patient characteristics and types and durations of the surgeries were similar between the groups (Table 1).

The visual analog scale (VAS) scores for each timepoint were not significantly different between the ESPB and PVB groups. The higher limit of the 95% CI for this difference (1.428 at PACU, 1.052 at 24 h, 1.176 at 48 h) was within the predefined noninferiority margin of 2 (delta). The total doses of rescue analgesics (110.24 ± 103.64 vs. 118.40 ± 93.52 MME, *p* = 0.767), the number of rescue analgesic events (5.88 ± 1.56 vs. 5.50 ± 1.45, *p* = 0.361), and satisfaction scores (3.5 (3.0, 4.0) vs. 4.0 (3.0, 5.0), *p* = 0.227) were not significantly different between the two groups (Figure 1). There were no significant differences in the intraoperative dose of remifentanil or the frequency of hypotension, bradycardia, or pleural puncture that occurred during the operation after the block (Table 2). The number of patients with moderate (VAS > 3) or severe (VAS > 6) pain was also similar between the two groups. We continuously identified the needle-tip position with ultrasound during the block. However, in the PVB-ICNB group, we recognized the pleural puncture, which was confirmed by the spread of the local analgesics into the pleura in two cases. The placement of the chest tube after the VATS procedure was performed as standard of care [11]. In both cases, the chest tube was removed and discharged without pneumothorax or other complications.

The hemodynamic data, including heart rate and mean arterial blood pressure during surgery, were also similar between the groups (Figure 2).

## 4. Discussion

The results of the present trial suggest that ESPB was not inferior in analgesic efficacy to PVB when combined with ICNB for attenuating surgery-related pain. Moreover, this combination block was similar in terms of patient satisfaction, analgesic requirement, and the frequency of hemodynamic perturbations. In addition, although the difference was not significant, pleural puncture was absent in the ESPB-ICNB group but present in two cases in the PVB-ICNB group.

VATS is a minimally invasive procedure that results in minimal tissue trauma. Nevertheless, it also causes significant and intense acute pain, which may lead to post-thoracotomy pain syndrome [12]. The scope manipulation and use of rib retractors in VATS may cause intercostal nerve injury [13]. In addition, multiple muscle incisions and pleural irritation from chest tubes can cause moderate to severe pain [1].

TEB has long been the standard procedure for pain control after thoracotomy. However, side effects such as hypotension, bradycardia, and pruritus are common, and analgesic failure often occurs because of an incorrect target position. In addition, catastrophic side effects, such as epidural hematoma, require close attention [14]. A previous meta-analysis showed that PVB has an equipotent analgesic effect after thoracotomy compared with TEB [15]. However, the paravertebral space is narrow, and pleural puncture risks are possible [16].

ESPB involves the injection of local anesthetics into the fascial plane between the transverse process and erector spinae muscle [17]. In the case of VATS, the ports are positioned in the intercostal space of the mid-clavicular and post-scapular line, and an incision measuring 4–5 cm is positioned in the mid-axillary line [18]. Mostly, the intercostal nerves originating from the anterior rami of spinal nerves are responsible for sensory innervation [19]. However, since adjacent nerves branch out and perform various anastomoses with each other, sensory innervation is not well-associated with the segment level [20]. Therefore, it is important that the analgesics spread not only to the pathway of the intercostal nerve but also to adjacent segments. A previous literature review reported that the injectates spread to the ventral rami in 13 out of 16 cadaveric studies and to the paravertebral space in 12 studies through the thoracic ESPB. In all 16 studies, craniocaudal spread over three levels was observed [21]. In previous analyses, ESPB showed comparable efficacy to PVB in terms of opioid consumption and pain scores [22,23]. Controversial results indicating that PVB is superior to ESPB in terms of pain scores and opioid consumption have been reported [7,24].

Although there is no consensus on the analgesic effect, ESPB has several advantages over PVB. Firstly, it is technically easier to contact the transverse process than it is to fix a needle tip in the relatively narrow paravertebral space. Therefore, high skill is not required for the practitioners, and the difference in efficacy was shown to be little among the practitioners [19]. Moreover, since the paravertebral space is adjacent to the pleura, the puncture risk in PVB is higher than that in ESPB [19]. Thirdly, in terms of anticoagulation, ESPB has an advantage over PVB, which is considered a neuraxial block. Furthermore, the paravertebral space is a noncompressible area, but the target of the ESPB is compressible. However, unpredictable spread of injectates in ESPB was reported in some trials [25,26]. Therefore, the use of ICNB in conjunction with ESPB may enhance the analgesic effect. ICNB can be easily performed without complications while directly observing the spread. However, PVB provides better analgesic efficacy than ICNB [27]. In the present study, single ESPB combined with ICNB was not inferior to PVB combined with ICNB in terms of analgesic efficacy and opioid consumption. This combination block can be an effective and safe option to control pain after thoracoscopic surgery.

Our study had several limitations. Firstly, the analgesic effect was compared using a subjective index. However, there is no objective indicator that measures the pain index, and several previous trials have used the NRS as an effective tool for evaluating analgesic efficacy [28,29]. Secondly, we did not study patients for a long duration. According to the literature, moderate to severe pain is maintained postoperatively in approximately 10% of patients at 52 weeks [30]. Therefore, if we evaluated the patients for a longer period, the analgesic efficacy between the groups might have been better exhibited. Thirdly, our single-center setting and small sample size may have been insufficient for validating the secondary endpoints of the study. Fourthly, since both procedures were performed under general anesthesia, examination of the dermatomal level for analgesia, such as the pinprick test, could not be performed. Fifthly, the total dose and number of rescue analgesics exceeded those commonly used in VATS. This may have been a confounding factor in comparing the efficacy between the groups. However, in the previous study, 56% of patients complained of moderate to severe pain when only intravenous PCA was used [31]. Reducing the prevalence of moderate to severe pain and the VAS pain score in the present study showed the advantage of the combined block in both groups. Lastly, we did not include a control group of patients that only received ICNB. Therefore, it is difficult to determine to what extent the combination of regional block techniques in VATS is really advantageous.

In conclusion, the present study is the first trial to compare the effects of ESPB and PVB with ICNB. Both groups provided an adequate analgesic effect in VATS. However, compared to PVB, ESPB is easier to implement and has advantages in terms of safety due to no adjacent vulnerable structures. The present study provided evidence that ESPB with ICNB may be an efficacious analgesic option in VATS.

## Figures and Tables

**Figure 1 jcm-11-05452-f001:**
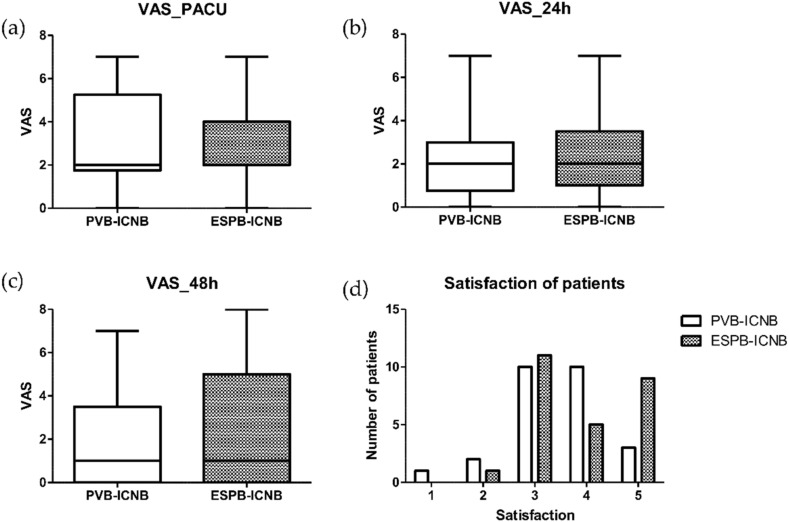
Visual analog scale (VAS) and satisfaction of patients: (**a**) VAS at PACU; (**b**) VAS at 24 h postoperatively; (**c**) VAS at 48 h postoperatively; (**d**) Satisfaction scores of patients. PACU: postanesthetic care unit, PVB: paravertebral block, ESPB: erector spinae plane block, ICNB: intercostal nerve block.

**Figure 2 jcm-11-05452-f002:**
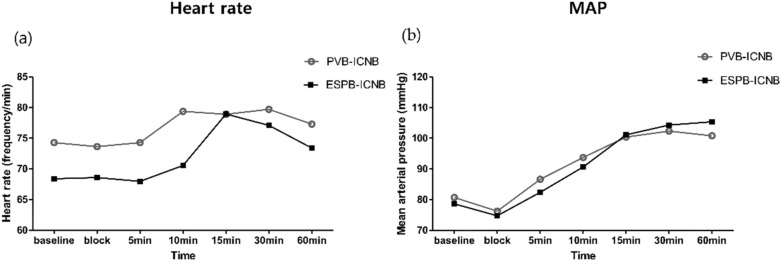
Hemodynamic data: (**a**) Heart rate; (**b**) Mean arterial pressure.

**Table 1 jcm-11-05452-t001:** Patient characteristics and surgery details.

	PVB-ICNB (*n* = 26)	ESPB-ICNB (*n* = 26)	*p*-Value
Female (%)	9 (34.6)	13 (50.0)	
Age	62.42 ± 13.11	60.31 ± 15.43	0.597
Height	161.09 ± 11.33	161.01 ± 10.14	0.978
Weight	62.34 ± 10.43	60.18 ± 10.72	0.464
BSA	1.66 ± 0.18	1.63 ± 0.17	0.591
Diabetes mellitus	11 (42.3)	6 (23.1)	0.143
Preoperative analgesics	3 (11.5)	3 (11.5)	1.000
Operation Type			
Lobectomy	12 (46.2)	13 (50.0)	
Segmentectomy	3 (11.5)	0 (0.0)	
Wedge resection	10 (38.5)	13 (50.0)	
Mediastinal mass	1 (3.8)	0 (0.0)	
OP time (min)	85.96 ± 48.48	97.69 ± 55.38	0.420

Values are displayed as the mean ± SD or *n* (%). PVB: paravertebral block, ESPB: erector spinae plane block, ICNB: intercostal nerve block, BSA: body surface area, OP: operation.

**Table 2 jcm-11-05452-t002:** Outcome measures.

	PVB-ICNB (*n* = 26)	ESPB-ICNB (*n* = 26)	95% CI	*p*-Value
Primary Endpoint				
VAS PACU	2.0 (1.8, 5.3)	2.0 (2.0, 4.0)	(−0.890, 1.428)	0.970
VAS 24 h	2.0 (0.8, 3.0)	2.0 (1.0, 3.5)	(−1.283, 1.052)	0.993
VAS 48 h	1.0 (0.0, 3.5)	1.0 (0.0, 5.0)	(−1.637, 1.176)	0.985
Above Moderate Pain (VAS > 3)				
VAS PACU	8 (30.8)	7 (26.9)		0.762
VAS 24 h	5 (19.2)	6 (23.1)		0.737
VAS 48 h	6 (23.1)	8 (30.8)		0.536
Above Severe Pain (VAS > 6)				
VAS PACU	4 (15.4)	2 (7.7)		0.390
VAS 24 h	1 (3.8)	2 (7.7)		0.556
VAS 48 h	1 (3.8)	2 (7.7)		0.556
Secondary Endpoints				
Rescue Analgesics (MME)	110.24 ± 103.64	118.40 ± 93.52		0.767
Number of Rescue Analgesic Events	5.88 ± 1.56	5.50 ± 1.45		0.361
Satisfaction of Patients	3.5 (3.0, 4.0)	4.0 (3.0, 5.0)		0.227
Remifentanil (µg)	511.62 ± 205.51	547.42 ± 224.35		0.551
Antiemetics				
Dose	1.59 ± 0.63	1.44 ± 0.67		0.408
Hypotension	2 (7.7)	4 (15.4)		0.390
Bradycardia	0 (0)	2 (3.8)		0.153
Pleural Puncture	2 (3.8)	0 (0)		0.153
Hospital Day	9.04 ± 4.20	9.27 ± 3.77		0.836

Values are displayed as the mean ± SD, *n* (%), or median (interquartile range). The pain score was assessed using a visual analog scale (VAS) (0 = no pain, 10 = worst imaginable pain). The rescue analgesic requirement was calculated in morphine milligram equivalents (MME). The satisfaction score was assessed using a five-point numerical scale (1 = extremely dissatisfied, 5 = extremely satisfied). VAS: visual analog scale, PACU: postanesthetic care unit, PVB: paravertebral block, ESPB: erector spinae plane block, ICNB: intercostal nerve block, SD: standard deviation, CI: confidence interval.

## Data Availability

The data presented in this study are available upon request from the corresponding author.

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
