# Peer review of "The Analgesic Efficacy of the Single Erector Spinae Plane Block with Intercostal Nerve Block Is Not Inferior to That of the Thoracic Paravertebral Block with Intercostal Nerve Block in Video-Assisted Thoracic Surgery"

_jcm, 2022, doi:10.3390/jcm11185452_

Round 1
Reviewer 1 Report
This is a randomized controlled trial that compares the effect of ESPB+ICNB and TPVB+ICNB. I have some questions as follow:
Methods
‘The BIS (Bisphectal 67 index) score was maintained between 40 and 60 through the control of remifentanil (0.03–68 0.08 mcg/kg/min) and sevoflurane or desflurane.’ Did you adjust remifentanil doses according to BIS levels?
I understand to administer a systemic analgesic, you waited the patient to feel pain. I have concerns with this analgesic algorithm as thoracic surgeries are associated with severe pain.
Perioperative management: there is no need to mention ICN block as there is a separate section for ICN block.
I have questions with the sample size calculation. Referred study was conducted in breast surgeries and ICN block was not performed.
Results
There were no drop-outs in a study that was conducted in thoracic surgeries. I congratulate the authors for their patient management.
‘The total doses of rescue analgesics (257.97±118.00 vs. 286.25±115.71 144 MME, p=0.387)’ I ask the authors how many times rescue analgesics (opioids and NSAIDs) were administered? How many times patients felt severe pain in this study (VAS>6)?
In the methods section, it was written that NRS scores were compared. Results (and abstract) include data of VAS scores.
Discussion
Mechanisms of TPVB and ESPB were well discussed. However, I would like to ask the authors why these blocks were combined with ICNB, and is it necessary to combine ICNB with TPVB?
Line 183: In this study, TPV block was performed at the end of the surgery and pleural puncture was occurred in 2 patients. However, in the results section ‘during surgery, a chest tube was placed in all patients so that pleural puncture would not induce pneumothorax’ was written. Please revise this statement to prevent the contradiction and discuss these 2 cases further.
Reviewer 2 Report
Thank you for the possibility to review the manuscript: „The analgesic efficacy of the single erector spinae plane block with intercostal nerve block is not inferior to that of the thoracic paravertebral block with intercostal nerve block in video-assisted thoracic surgery: A randomized controlled trial“.
The article is quite well written, but I have some comments on it.
Major concerns:
- I am concerned that primarily the design of this study was not chosen appropriately. In general, the VAS of the patients is very low in both groups (median around 2) - the pain in both groups was therefore very well controlled and this may be the reason why there was no difference between the groups. Isn't the combination of regional technique with local application (ICNB) and the administration of non-steroidal analgesics excessively and unnecessarily generous for VAST? It would be ideal to include a third group of patients who will only receive ICNB. I understand that this is probably not possible now. So, could the authors at least comment more on whether there are existing data showing a clear benefit of the combination of ICNB+PVB or ICNB+ESBP compared to PVB alone?
- In the tab no. 2 in the section "above moderate pain" it is not explained what the given values mean. How many patients had the VAS>3 ?? If this was a significant group of patients, would it not be better to use the mean rather than the median for evaluation? Doesn't the median underestimate the final result and therefore doesn't it hide the difference in both groups? I kindly ask the authors to provide the mean +/- SD and p values for both groups at least as the part of the answer to my review.
Minor concerns:
- The title is quite long and confusing. I recommend at least omitting "a randomized controlled trial" if the editor also agrees.
- On the contrary, in the abstract and in the introduction, I would state what kind of study it is - monocentric, randomized...
- Citations are missing in several places in the text - line: 33 (“…and morbidity”), 204 (…among the practitioners), 205 (…that in ESPB).
- The methods lacks information on how patients were randomized. Was the study blinded? Did the patients know which block was performed on them? Did the people conducting the study know: the nurses, the attending PACU physician, the researcher evaluating the data?
- Somewhere in the text “VAS” is mentioned and somewhere “NRS”. Please unify this or explain the differences and when was each method used.
- How was the value of the non-inferiority margin (=2) determined?
- I recommend to omit the figure 1. Since no patients were excluded, the figure is meaningless. It will be sufficient if the authors provide this information in the text.
- In Table 1, I would add the information whether any of the patients had chronic pain or analgesic therapy before the operation.
- It would also be good to state what comorbidities the patients had. Above all, diabetes mellitus can affect the perception of pain.
- The sentence “During surgery…” (line 149-150) will fit better in methods section.
- How was the pleural puncture (as a complication of peripheral block) diagnosed when all patients had already the chest tube inserted??
- In the conclusion, it would be appropriate to state what is the opinion of the authors? What combination of techniques is the best for the patient in terms of pain control, risk of complications, difficulty of execution.
- The sentence starting at line 207 (Therefore, if ESPB…) is hard to understand, please reword it.
Round 2
Reviewer 1 Report
There are no further issues.
Author Response
Thank you for your valuable comments.
Reviewer 2 Report
The quality of the manuscript has increased a bit. There are still a few things that the authors should change...
- Describe the type of the study better, e.g.: monocentric single-blinded (or double-blinded?) randomized controlled non-inferiority trial.
- What the formulation "Rescue analgesics number" means ? Is it the number of patients who had to receive fentanyl dose or the number of events of fentanyl administration ? Please specify in the table and the text.
- When the authors omitted the patient enrollment flow diagram, they should at least include the information in the methodology text as had been recommended.
- In the limitations of the study, I recommend mentioning that the study did not include a control group that would only receive ICNB, and therefore it is difficult to determine to what extent the combination of regional techniques in VATS is really advantageous.
- I recommend to omit the sentence: "Nevertheless, to the best of our knowledge, the present trial is the first to compare the analgesic effects of ESPB and PVB combined with ICNB." It is repeated again in conclusion.
- Please, check the English again in final version, there are a few mistakes - mostly in new sentences.
